# Best-Evidence for the Rehabilitation of Chronic Pain Part 1: Pediatric Pain

**DOI:** 10.3390/jcm8091267

**Published:** 2019-08-21

**Authors:** Lauren E. Harrison, Joshua W. Pate, Patricia A. Richardson, Kelly Ickmans, Rikard K. Wicksell, Laura E. Simons

**Affiliations:** 1Department of Anesthesiology, Perioperative, and Pain Medicine, Stanford University School of Medicine, Stanford, CA 94304, USA; 2Faculty of Medicine and Health Sciences, Macquarie University, Sydney, NSW 2109, Australia; 3Research Foundation-Flanders (FWO), 1000 Brussels, Belgium; 4Department of Physiotherapy, Human Physiology and Anatomy (KIMA), Faculty of Physical Education & Physiotherapy, Vrije Universiteit Brussel, 1090 Brussels, Belgium; 5Pain in Motion International Research Group, 1090 Brussels, Belgium; 6Department of Physical Medicine and Physiotherapy, University Hospital Brussels, 1090 Brussels, Belgium; 7Department of Clinical Neuroscience, Psychology division, Karolinska Institutet, 171 65 Stockholm, Sweden

**Keywords:** chronic pain, children pain rehabilitation, best evidence

## Abstract

Chronic pain is a prevalent and persistent problem in middle childhood and adolescence. The biopsychosocial model of pain, which accounts for the complex interplay of the biological, psychological, social, and environmental factors that contribute to and maintain pain symptoms and related disability has guided our understanding and treatment of pediatric pain. Consequently, many interventions for chronic pain are within the realm of rehabilitation, based on the premise that behavior has a broad and central role in pain management. These treatments are typically delivered by one or more providers in medicine, nursing, psychology, physical therapy, and/or occupational therapy. Current data suggest that multidisciplinary treatment is important, with intensive interdisciplinary pain rehabilitation (IIPT) being effective at reducing disability for patients with high levels of functional disability. The following review describes the current state of the art of rehabilitation approaches to treat persistent pain in children and adolescents. Several emerging areas of interventions are also highlighted to guide future research and clinical practice.

## 1. Introduction

Chronic pain is a prevalent problem among children and adolescents, with epidemiological data indicating approximately 30% of children experience pain persisting for 3 months or longer [1,2]. The most common pain complaints in children include migraine, abdominal pain, and musculoskeletal pain [1]. The presence of chronic pain has a significant negative impact on functioning, with impairments across academic, social, and recreational domains, as well as family functioning [3]. Given this broad impact, treatment for chronic pain typically focuses on functional improvements across domains.

Rehabilitation for chronic pain applies the biopsychosocial model [4], which accounts for the complex interplay of the biological, psychological, social, and environmental factors that contribute to and maintain pain symptoms and related disability. Most interventions for chronic pain are within the realm of rehabilitation, based on the premise that behavior has a broad and central role in pain management. These treatments are typically delivered by one or more providers in medicine, nursing, psychology, physical therapy, and/or occupational therapy. In chronic pain management, as contrasted with acute pain treatment, the emphasis shifts from immediate analgesia to functional improvements in the presence of pain [5,6]. Given this, this review will focus solely on rehabilitation interventions, including psychological, behavioral, and physiological interventions.

Given all of the domains involved and impacted by chronic pain, treatment typically requires a comprehensive and multidisciplinary approach, most often achieved through psychological interventions, as well as physical and occupational therapies [7]. Multidisciplinary teams often consist of providers from several specialties who work together to develop a treatment plan for the patient and family [8]. An interdisciplinary model, comprised of the same specialists, differs slightly from a multidisciplinary team as the team members work together in a more fluid way (often housed within the same institution or clinic), with more extensive collaboration and shared treatment goals [8].

Rehabilitative treatments are also delivered across several levels of care, including outpatient, intensive outpatient (i.e., day treatment), and inpatient. Often thought of as the first level of care, outpatient interventions can be collaborative and multidisciplinary (e.g., if a psychologist and physical therapist were housed within the same medical system and occasionally corresponded regarding patient progress). However, outpatient interventions are typically delivered independent of each other. Unfortunately, adherence to treatment recommendations in outpatient pain clinics is often suboptimal [9], which underscores the importance of thorough assessment and delivery of tailored treatment approaches that best meet the individual needs of the child and family. Furthermore, patients with more severe pain-related disability or who have been unsuccessful in outpatient therapies may require more comprehensive and intensive interventions [8].

This review has several aims. First, we present an overview of the state-of-the-art of rehabilitative interventions for children and adolescents with chronic pain. In order to present the best evidence in rehabilitation for pediatric chronic pain, this review primarily relies on systematic reviews and meta-analyses. However, to incorporate the most recent evidence, methodologically sound clinical trials (e.g., randomized controlled trials, sample size > 20, clearly described interventions) that have not yet been integrated into the available reviews are included as well. For this review, a non-systematic search of the literature was performed in PubMed and Google Scholar using the following search strategy: ((child OR pediatric) OR adolescent AND (chronic pain (Text Word)) AND rehabilitation (Text Word)). The inclusion of chronic pain and rehabilitation terms resulted in a search that pulled for the primary rehabilitative interventions used to treat pediatric chronic pain. All titles and abstracts (total of 399) of articles were then separately screened by two authors (L.E.H., J.W.P.) for inclusion. Our second aim is to inform clinicians of innovative and emerging treatments that can potentially enhance treatment delivery and patient outcomes. Lastly, this review intends to identify evidence gaps among interventions that warrant further study, and to serve clinical researchers to build upon the best evidence for designing future trials, implementing studies, and developing innovative future studies.

## 2. State of the Art of Rehabilitation for Pediatric Chronic Pain

There is strong evidence to support early, targeted psychological and physiological intervention for pediatric chronic pain, with most approaches sharing common features: Pain education, psychological interventions, and physical and occupational therapies [7]. Psychological interventions for pediatric chronic pain focus on the self-management of pain and disability, with the ultimate goal of a return to baseline functioning [10,11]. Components of psychological interventions for chronic pain include, but are not limited to, psychoeducation, relaxation training, identifying and addressing negative cognitions, acceptance and values-based exercises, behavioral exposures, and parent coaching [12]. There is substantial evidence suggesting these interventions are effective at reducing pain severity, disability, and psychological distress (e.g., anxiety) in children with chronic pain [13,14,15]. Within pain conditions, psychological interventions have been found to effectively reduce pain in headache, abdominal pain, and musculoskeletal conditions, and functional disability in abdominal and musculoskeletal pain [14].

Physiological and rehabilitative interventions for pediatric chronic pain, including physical and occupational therapy, focus on improving physical functioning by progressively engaging children in previously avoided activities and taking a self-management approach to pain [16,17]. The goal of these interventions is to improve strength, flexibility, endurance, joint stability, tolerance for weight-bearing, coordination, balance, and proprioception [5,18,19,20]. Because the goal of these therapies is to promote independence (i.e., ability to manage daily life without excessive support from parents and caregivers) and a return to functioning (e.g., return to school and sport), active interventions (e.g., exercise) have a more significant role than passive interventions (e.g., massage or transcutaneous electrical nerve stimulation (TENS)) [21]. The goals of physical and occupational therapeutic interventions are often focused on independent functioning, as well as improved coping and increased self-efficacy, as opposed to pain reduction [22,23]. Following a thorough assessment within a biopsychosocial framework, including an assessment of functional goals, a developmentally-appropriate and individualized therapeutic treatment plan is developed and implemented [24]. The following section thoroughly reviews the evidence of effectiveness of the aforementioned rehabilitative treatments across outpatient, as well as intensive outpatient and inpatient treatment programs (see Table 1).

### 2.1. Pain Related Education

The goal of psychoeducation is to provide the child and family with an explanation of the differences between acute and chronic pain and to emphasize the non-protective nature of chronic pain [12]. Psychoeducation is typically guided by the biopsychosocial model [4] and is an important component of treatment as it provides the family with a rationale as to how psychological interventions can be effective in addressing pain and associated disability [12]. Although psychoeducation is typically embedded in any comprehensive cognitive-behavioral treatment package, the clinicians and researchers in the field of physiotherapy have delved deeper into educating patients about pain science as a therapeutic tool and have worked to test the efficacy of pain science education both as a specific treatment component [26,58], as well as in combination with other biopsychosocially-oriented treatment components [59,60,61] Indeed, adding a cognition-targeted active approach to pain science (e.g., progression to the next phase of education is preceded by an intermediate phase of imagery or work on cognitions that might hinder progression) is considered critical in achieving larger long-term therapy effects, given that pain science as a stand-alone treatment only demonstrates small to medium effect sizes [62,63]. Although already used in clinical practice worldwide, research on pain science education in the pediatric pain field is just beginning.

#### Pain Science Education

“Pain science education”, also called “pain neuroscience education” [63,64], “therapeutic neuroscience education” [65,66], or “explaining pain” [26], aims to change one’s conceptual understanding of pain [67]. To enhance rehabilitation treatments, pain science education provides a foundation for understanding principles that guide biopsychosocial interventions for persistent pain [35]. Pain science education teaches people about the underlying biopsychosocial mechanisms of pain, including how the brain produces pain and that pain is often present without, or disproportionate to, tissue damage. In more complex and persistent pain states, this also includes peripheral and central sensitization, facilitation and inhibition, neuroplasticity, immune and endocrine changes [58]. Evidence shows that understanding pain decreases its threat value which, in turn, leads to more effective pain coping strategies [61,68]. Given that children with chronic pain often have significant problems with functioning (e.g., more school absenteeism and lower participation in daily, after-school, and family activities) contributing to lower quality of life, less physical fitness, and eventually more pain [69], pain science education may prepare and prime children with chronic pain for biopsychosocial treatments.

Pain science education is commonly part of multidisciplinary pain treatment [70] and can utilize freely available online resources [25] that complement pain science education/communication which is typically individually-tailored and thereby primarily delivered by a therapist. Additionally, PNE4Kids, a pain science curriculum for children (6–12 years old), has recently been developed [27] and is freely available for clinicians at http://www.paininmotion.be/pne4kids. Although there is meta-analytic evidence for adults suggesting that pain science education improves outcomes [71], evidence in pediatric chronic pain is scarce but promising with Andias et al. [72] providing support of a combined approach (pain science education + exercise therapy) in adolescents with chronic idiopathic neck pain, with data showing that this type of intervention is feasible and beneficial in pediatric patients with chronic pain. Yet, some methodological shortcomings were present in this study, such as the rather small sample size (*n* = 43) and the control group who did not receive any treatment (nor attention from the therapists). Therefore, further methodologically sound research is needed to assess both conceptual and behavioral change in relation to pain science education.

### 2.2. Physiological Self-Regulation Training

An often recommended intervention for children with chronic pain is training in self-regulation of physiological responses to pain (e.g., heart rate, breathing rate, skin temperature, and muscle tension). Relaxation-based strategies typically include deep-breathing exercises, progressive muscle relaxation, and imagery [10]. Studies have shown the direct benefits of relaxation techniques for children with persistent pain including slowing of heart rate and breathing, increased blood flow to the muscles, and decreased muscle tension e.g., [73]. These bodily changes have also been found to reduce the experience of stress and anxiety [74]. Often used in conjunction with relaxation training, biofeedback provides real-time feedback to the child related to the physiological processes and changes (e.g., changes in heart rate or skin temperature) that occur in the body when engaging in aforementioned relaxation techniques [29]. There is also some evidence for self-hypnosis. Several studies highlight the utility of self-hypnosis with pediatric procedural pain [30,31] and there is some evidence that it may be helpful with chronic conditions as well [32]. Examining a sample of 300 children with functional abdominal pain, Anbar [75] found that 80% of patients demonstrated improvement in pain following a course of self-hypnosis. Another study examined the efficacy of self-hypnosis in 26 children with chronic pain. Results demonstrated that self-hypnosis was significantly associated with decreased pain intensity, as well as improvements in functioning across academic and social domains and sleep [76].

#### Mindfulness-Based Stress Reduction and Yoga

Mindfulness-based stress reduction (MBSR) involves teaching patients mindfulness and focuses on bringing attention to the present moment, with the thought that shifting attention to the present allows for the use of positive coping strategies [77]. Data from pilot trials demonstrates evidence for the efficacy of MBSR for reduction of pain and stress, with improvements maintained at 6-month follow-up [35,78]. Additionally, there is some evidence for the role of yoga [34] and massage [79,80] in treating pediatric chronic pain. Integrating aspects of MBSR and yoga might help enhance interventions delivered within this population.

### 2.3. Cognitive Skills Training

Cognitive skills training focuses on the identification of negative and maladaptive thoughts/cognitions with the goal of systematically reframing and changing these thoughts [81]. Several clinical trials have demonstrated that children with chronic pain benefit from cognitive skills training [15]. Cognitive techniques such as cognitive restructuring, problem-solving, and positive self-talk, have been shown to be effective techniques for reducing negative thoughts associated with pain and related disability [10,13,14].

Incorporating components of Acceptance and Commitment Therapy (ACT) [82,83] into treatment may be beneficial, and there is evidence to suggest that ACT can be effective for children with chronic pain [39,84]. ACT focuses on increasing psychological flexibility and engagement in valued activities (e.g., willingness to go to school or to a friend’s house even if pain is present) [82,85]. ACT differs from traditional cognitive therapy in that it focuses on changing the relationship the child has with distressing and negative thoughts as opposed to changing the thoughts themselves. This is done through exercises focused on cognitive defusion, which aims to increase the child’s ability to notice the thought and how it influences behavior, rather than changing the content of the thought [86]. One study examining acceptance and values-based treatment for adolescents with chronic pain found that adolescents improved in self-reported functioning, as well as on objective measures of physical performance and reported a decrease in anxiety and catastrophizing [84] following intervention. Additionally, Wicksell and colleagues [85] examined mediators of change in ACT and found that ACT worked through improvements in processes related to psychological flexibility rather than through changes in traditional CBT constructs, providing additional evidence that ACT may be functionally different from traditional cognitive-behavioral treatments.

### 2.4. Behavioral Exposure

Operant-behavioral theories have long been applied to chronic pain populations to understand the association between pain severity and pain-related disability [87,88]. The Fear Avoidance Model [89,90] describes how heightened fear of pain and continued avoidance of activities that might exacerbate pain leads to prolonged disability, and recent work has focused on pain-related fear in children and adolescents and the application of the Fear Avoidance Model of Chronic Pain within pediatric patients [91,92]. Exposure-based treatments for pediatric chronic pain aim to improve functioning by exposing patients to activities they are currently avoiding due to fear of pain. In a study examining the efficacy of behavioral exposure within an ACT framework for children and adolescents with chronic pain, results demonstrated greater reductions in pain severity, functional disability, and fear of pain for patients who received the exposure treatment compared to those who received standard multidisciplinary treatment [39].

Graded in-vivo exposure, a treatment typically delivered by a psychologist and physio or occupational therapist [93,94], thus considered an interdisciplinary outpatient treatment, is now being evaluated in children and adolescents with chronic pain (described further in the Interdisciplinary Outpatient Pain Treatment section) [51]. The first single case experimental design (SCED) trial of graded in-vivo exposure in youth demonstrated robust improvements in pain-related avoidance and pain intensity with increased activity engagement at the end of treatment with decreases in pain-related fear and catastrophizing observed at 3-month follow-up with improvements across outcomes maintained at 6-month follow-up (Simons et al., [95]). Additionally, work has been done to incorporate interoceptive components, which involve having the child imagine increases in pain severity, into exposure treatments for children with chronic pain [96,97]. Use of interoceptive exposure techniques have been associated with decreased pain intensity and school avoidance, and data suggest that using these techniques are beneficial at reducing pain severity and altering relevant emotions related to pain [96].

### 2.5. Parent Coaching

While the aforementioned interventions often consider the child to be the primary treatment target, parents play a critical role in managing pain and maintaining or improving functioning [10]. At the very basic level, parents are often taught the relaxation and cognitive skills along with the child so that they are better able to help their child carry out the interventions at home [10]. Treatment with parents also focuses on shifting parent attention and behavioral responding toward encouraging function in the presence of persistent pain, while coaching the child to use coping skills to support functioning (i.e., contingency management). Findings from a systematic review of parent–child interventions, including cognitive-behavioral and family-focused treatments, found that these interventions could be beneficial in improving parent behaviors, e.g., reducing attention to pain symptoms, encourage functioning despite pain [36].

There is also evidence to suggest that parents of children with chronic pain experience significant emotional distress related to their child’s pain (e.g., [98]). Recent work has been done to adapt problem-solving skills training (PSST), which teaches parents structured approaches to solving problems and targets parent distress, with results demonstrating that psychological interventions focused on reducing parent distress were effective [41,42]. Furthermore, a recent review [99] found that psychological interventions also improved parent mental health across a chronic illness sample, including parents of children with chronic pain. These findings support the notion that parent distress impacts child functioning [100]. Therefore, it is critical to consider parent distress when working with this population, as accurate assessment and treatment of parent distress, in addition to behavioral functioning, may have important implications for child outcomes. Additionally, there are several books written for parents that provide support and instruction on how to implement the aforementioned strategies and support coping and functioning in their children [44,101,102,103].

### 2.6. Physical Therapy

When working with children and adolescents with chronic pain, the key objectives of a physical therapist include encouraging the adoption of regular exercise, facilitating repeated exposure to movement in the presence of pain, and educating families regarding misconceptions about anatomy, physiology, pain, exercise and activity [48]. To assist the child in achieving functional goals, physical therapy works to improve strength, flexibility, endurance, joint stability, tolerance for weight-bearing, coordination, balance, and proprioception [5,18,19,20].

Exercise is a crucial component of rehabilitation for children and adolescents with chronic pain [7] and there are data to suggest that earlier experience with exercise is associated with better adherence [9]. Exercise activities for pain in the lower extremities may focus on jumping, fast-paced walking and/or running, climbing stairs, balance and coordination activities, and age-appropriate physical education activities and sport drills, whereas upper extremity exercises typically focus on strengthening and coordination drills [104]. It is important that exercises occur in a variety of settings, such as in the gym with equipment, at home without equipment, in a pool, or out in public settings, to assist with generalization of skills and reduce site-specific exercise behavior [105]. It is also important to take a behavioral management approach when increasing physical activity, most often achieved through gradual exposure to activities and pacing, which involves increasing intensity gradually as tolerance builds [19,48].

### 2.7. Occupational Therapy

Another vital component of rehabilitation for chronic pain is occupational therapy [8]. Occupational therapy differs from physical therapy in that the focus of interventions are on maximizing independence in age-appropriate activities of daily living and self-care (e.g., bathing, dressing, grooming), as well as academic (e.g., handwriting) and family activities (e.g., participation in chores) [17,19,48]. These goals are often achieved through individualized strategies such as psychoeducation, participation in games (e.g., standing up while playing a board game, or participating in games that requiring reaching or bending-designed based on patients functional goals), sensory discrimination, and developing a daily schedule to support engagement in meaningful activities throughout the day [5,8,106]. Desensitization, a technique used to reduce physical sensitivity to certain stimuli, is another important intervention provided though occupational therapy, particularly for patients with central pain sensitization (e.g., Complex Regional Pain Syndrome; CRPS) who experience difficulties tolerating physical stimulations and sensations on affected areas of the body. These patients may guard or protect the sensitive area in an attempt to avoid it being touched. In severe cases, patients may be unable to tolerate pressure from clothing items, such as socks, shoes, and tighter pants. To address this, the occupational therapist engages the patient in desensitization exercises, which may include rubbing the sensitive area with various textures including tissue, feather, textured fabrics, and towels, to gradually expose the nervous system to different sensations with the goal of retraining the brain to process these stimulations more typically [104].

An early meta-analytic review [107] found that conventional (i.e., monodisciplinary) physical and occupational therapy are better than no treatment or only medical treatment. There are also data to suggest that for specific conditions, such as CRPS, early individualized intensive physical therapy is considered best [108,109]. However, monodisciplinary rehabilitation treatments have been found to be inferior to multidisciplinary and interdisciplinary treatment approaches, where physical and occupation therapy are combined with psychological intervention [19,107].

### 2.8. Addressing Comorbidities

Sleep is an important aspect of health and development in childhood and adolescence. Unfortunately, disturbances to sleep, including insomnia and delayed sleep phase, are prevalent in children with chronic pain [110] and are associated with negative emotional, cognitive, and behavioral consequences [111]. Thus, thorough assessment of sleep and sleep hygiene (i.e., habits that might affect sleep onset or maintenance throughout the night, such as consumption of caffeine, spending too much time during the day in bed, use of electronics at bedtime) is warranted [10]. There is evidence to suggest psychological interventions are effective at addressing sleep problems. Specifically, cognitive behavioral therapy for insomnia (CBT-I) has been found to be effective for adolescents with co-occurring physical and psychological conditions, including adolescents with chronic pain [40]. Outcomes from the pilot trial demonstrated improvements in insomnia, as well as improvements in sleep quality and sleep hygiene, psychological symptoms, and overall health-related quality of life [40].

Additionally, chronic pain is often associated with comorbid psychiatric concerns including depression, anxiety disorders, and post-traumatic stress disorder (PTSD) [112] and the co-occurrence is likely bidirectional in nature. In other words, psychological symptoms could potentially be a contributing factor and an outcome of having chronic pain [33,113]. Additionally, there are data to suggest that chronic pain is a risk factor for suicidal ideation in adolescents, and clinicians should be alert to suicidal ideation and/or attempt within the population [114]. Consultation with and involvement of psychiatric care should be incorporated into treatment when appropriate. Further, assessing for adverse childhood events, trauma, or maltreatment may also be important and exposure to early childhood adversity may hinder the ability to effectively implement interventions. In addition to depression and anxiety, the presence of neurological and/or neuropsychiatric symptoms (e.g., conversion disorder) co-occur in pediatric pain populations [115]. Effective interventions need to target co-morbid mental health disorders and identify underlying mechanisms that serve to maintain mental health and pain conditions.

### 2.9. Interdisciplinary Outpatient Pain Treatment

Over the last several years, effort has been made to examination the efficacy of interdisciplinary interventions delivered at the outpatient level. Fibromyalgia Integrative Training for Teens (FIT Teens) combines cognitive-behavioral interventions with neuromuscular exercise training [49]. Results from the pilot randomized controlled trial (RCT) comparing FIT Teens to CBT-only demonstrated that adolescents who participated in FIT Teens experienced significant improvements in disability and greater decreases in pain intensity compared to the CBT-only condition, suggesting that FIT Teens provides additional benefits above and beyond CBT for children and adolescents with fibromyalgia. Additionally, there are several emerging interventions focusing on graded in-vivo exposure therapy (GET) for children and adolescents with chronic pain [51]; (GET Living, NCT: 03699007). One ongoing randomized controlled trial (RCT) in the Netherlands, “2B Active”, combines GET and physical therapy to increase functioning by having patients complete activity exposures [51]. Additionally, there is an ongoing RCT in the United States comparing GET Living to multidisciplinary pain management in children with chronic pain (GET Living, NCT: 03699007). Similar to 2B Active, GET Living utilizes a psychologist and a physical therapist to deliver exposure interventions. However, different from 2B Active, GET Living specifically targets pain-related fear and avoidance.

### 2.10. Intensive Interdisciplinary Pain Treatment

Often when patients are unsuccessful at outpatient treatments, a more intensive, interdisciplinary pain treatment (IIPT) is required [19]. There is evidence from a systematic review and meta-analysis to suggest that IIPT may be effective at reducing disability and maintaining this reduction after treatment for a subgroup of patients [19]. Specifically, children and adolescents have demonstrated improvements in pain intensity, pain-related disability, and symptoms of depression post-treatment, with improvements maintained at 3-month follow-up. To be considered an IIPT program, the program includes three or more disciplines housed within the same facility (e.g., pain specialist, psychologist, and physical therapist) who work in an integrated manner to provide treatment in a day hospital or an inpatient setting. IIPT programs can be day treatment or inpatient and typically require the patient to participate in exercise-based therapies (PT and OT) as well as psychological interventions, for a total of eight hours per day [19].

Eccleston and colleagues [116] were the first to examine the effectiveness of intensive interdisciplinary pain treatment. The program examined was a 3-week multidisciplinary treatment for patients and parents, with results demonstrating immediate improvements in functioning. After a 3-month follow-up, data showed a significant decrease in anxiety, pain catastrophizing, disability, and improvements in school attendance. Another study compared a 3–4 week intensive day hospital rehabilitation program with standard outpatient treatments, which included various combinations of medical treatment, psychological, and physical therapies [117]. While there were improvements noted across both treatment groups, patients enrolled in the intensive day rehabilitation program had significantly larger improvements in functional disability, pain-related fear, and willingness to adopt a self-management approach to treating pain [117]. A recent study [5] examined the effects of an intensive day treatment program in which patients completed 1–2 half day sessions per week (lasting approximately 4 h each) for 4–8 weeks. Results indicated improvements in pain severity, as well as physical and psychological functioning [5]. To date, there has only been one randomized control trial (RCT) comparing intensive interdisciplinary pain treatment (IIPT) to a waitlist control group [106]. The IIPT utilized in this trial was a manualized program consisting of 6 treatment modules including pain psychoeducation, pain coping skills, cognitive intervention to target emotional distress, family therapy, physiotherapy, and parent sessions. Immediate effects were achieved for pain-related disability, school attendance, depression, and catastrophizing, with pain intensity and anxiety decreasing at 6-month follow up [106]. These results are consistent with outcomes from other intensive interdisciplinary pain rehabilitation treatment centers [108,118].

While all IIPT programs share the primary goal of improved functioning across domains, there is variability across programs with regard to structure, organization, and frequency of treatment delivery across disciplines [8]. One major distinction is that of intensive outpatient and inpatient treatment models. In comparing outcomes reported from intensive outpatient [108] and inpatient programs [118,119], patients in each program demonstrate significant functional improvements. Of note, several intensive pain rehabilitation programs offer both inpatient and day hospital programs, with patients triaged to level of care based on individual needs [17]. To our knowledge, outcomes between levels of care within the same facility have yet to be published. Another difference between intensive pain programs includes length of stay. For example, some programs have a fixed, 3-week length of stay, while others have a more flexible length of stay which is often established based on individual patient needs. Despite these differences, significant functional improvements are reported for these treatment programs. Continued examination of outcomes within and between treatment programs is warranted, as is further examination of mechanisms of change for patients undergoing IIPT treatment.

A recent study conducted a cost-analysis of an interdisciplinary pediatric pain clinic by retrospectively reviewing billing data for inpatient admissions, emergency department, and outpatient visits and associated costs and reimbursements [120]. Data examined included healthcare costs for patients 1 year prior to initiating interdisciplinary services with costs 1 year after initiating services. Cost-analyses of pre-pain clinic costs found cost reductions 1 year post clinic participation (up to $36,228 to the hospital and $11,482 to insurance, per patient, per year), providing economic support for interdisciplinary intervention for children with chronic pain [120].

### 2.11. Emerging Pain Treatment Intervention Formats

**One-day workshops.** One day group-based psychological interventions for children with chronic pain have inherent benefits as it allows children and adolescents to meet peers with similar struggles and allows them to receive social support and benefit from shared experiences. These workshop programs can also be cost and time effective. One such program, The Comfort Ability [52] in an intensive one-day intervention, delivered concurrently to children and their parents, that introduces cognitive-behavioral skills of pain management and helps families develop a plan to support functional improvement. The workshop is currently available across 15 children’s hospitals in the United States and Canada. Preliminary evaluation of this workshop demonstrates improvements in child functioning, depressive symptoms, and pain catastrophizing, which persist at 1-month follow-up. Additionally, parents report improvements in responses to their child’s pain and beliefs regarding their child’s ability to manage pain [52].

**Internet and mobile applications.** Recently, effort has been made to address access barriers for pediatric pain management services. Palermo and colleagues developed an 8-week online psychological intervention for children and their parents (WebMAP). Online modules included relaxation training, cognitive strategies, parent operant techniques, communication strategies, and interventions focused on sleep and activity engagement. Pilot data demonstrated that internet-delivered pain management reduced barriers of access to care and was effective at reducing pain-related disability [121]. The program was further developed into a mobile app version with data also indicating greater reduction in pain intensity and functional disability post treatment compared to waitlist control [54]. Other mobile-based technologies have been developed to assist patients in remotely self-monitoring symptoms and deliver interventions involving goal-setting for improving functioning, coping skills training and practice, and social support via discussion boards, goal sharing, and group-based challenges [55]. Additionally, an ACT based digital intervention for individuals with chronic pain has recently been developed in a series of studies with desktop as well as mobile use [122]. Results from an RCT with adults (*n* = 113) showed moderate to large effects in primary and secondary outcomes, with effects remaining 12 months following the end of treatment. Additionally, a review examining remotely delivered psychological therapies found that they were beneficial at reducing pain intensity across pain groups [123]. While these programs allow for patients to have access to treatment, remotely-delivered interventions may not be appropriate for all patients, and more complex patients would likely benefit from more intensive treatments.

**Augmented reality and virtual reality.** Augmented reality and virtual reality (VR) have been found to be an effective tool for reducing pain sensations in patients with acute pain [56,57,124]. One recent study examined the effects of VR in patients with chronic right arm pain secondary to a diagnosis of complex regional pain syndrome (CRPS), type 1. Similar to results found within acute and procedural pain samples, Matamala-Gomez and colleagues [125] found that multisensory interventions that manipulated body from VR modulated pain perceptions. Continued research on the effectiveness of VR within the pain rehabilitation setting is needed and should be a focus of future research within this population.

### 2.12. Summary of Rehabilitative Treatments for Pediatric Chronic Pain

In sum, rehabilitation for pediatric chronic pain applies the biopsychosocial model, which takes into account the complex interplay of biological, psychological, social, and environmental factors that contribute to and maintain pain and related disability. Given all of the domains impacted by pain, rehabilitation typically require a comprehensive and multidisciplinary approach. Currently, there is strong evidence to support early, targeted, treatments, with most rehabilitative interventions including pain education, psychological interventions, and physical and occupational therapies. Promising directions for clinical practice and research are discussed below.

## 3. Promising Directions for Clinical Practice

Given the expansive growth of rehabilitation interventions for youth with chronic pain, it is imperative to match individual patients with the appropriate treatment modality and level of intensity. The use of a screening tool, such as the Pediatric Pain Screening Tool (PPST) [126] could potentially be used to facilitate efficient treatment allocation. PPST is a 9-item screening tool used to identify prognostic factors (e.g., sleep disturbance, depression, anxiety) associated with adverse outcomes, with allocation to the high-risk group based upon responses to psychosocial items. The PPST can be easily delivered within a busy clinical setting and allows providers to quickly and effectively identify medium to high risk youth who may benefit from access to more comprehensive, multidisciplinary treatments [126]. Administering the PPST would allow patients to be triaged to the appropriate level of care, without having to trial treatments that might not be appropriate given their level of risk. For example, a patient who screens medium to high risk could be triaged to initiate both psychology and physical therapy, whereas a low-risk patient might benefit from physical therapy alone. Efforts to better match individual patients with specific treatments might help to reduce “treatment failure” that some patients experience when they engage in treatments that poorly match their current symptoms and functioning (e.g., the need for CBT-I for sleep difficulties).

Further, attempts to tailor the interventions delivered within each discipline might also be beneficial. For example, when a patient is triaged to psychology for pain management, extra effort should be made by the provider to assess what specific treatment modality might be most beneficial. For example, a patient with musculoskeletal pain who is experiencing high pain-related fear and avoidance may benefit from including a more targeted graded exposure treatment approach, as opposed to solely focusing on historically popular components of cognitive-behavioral interventions for chronic pain (e.g., relaxation skills training). Lastly, continued effort should be made address barriers to access of care and continued effort should be made to integrate one-day workshops that can be delivered on the weekends e.g., the Comfort Ability [52], as well as internet-based and mobile application treatments [54,55]. See Figure 1 for visual overview.

## 4. Promising Directions for Research

Future research should focus on establishing clinical cut-off’s for measures assessing core outcome domains, as this will allow for better evaluation of clinically significant change post-treatment. Along these lines, it will be important to explore emerging treatment targets (e.g., assessing pain-related fear and avoidance vs. pain catastrophizing vs. functional disability, pre/post treatment). It will also be important to continue to examine innovative and targeted multidisciplinary treatments. Over the last several years, effort has been made to develop interdisciplinary outpatient treatments, such as FIT Teens [49], 2B Active [51], and GET Living (NCT: 03699007), and preliminary outcomes are promising [51].

There is also a need to explore the processes and mechanisms of change within pain rehabilitation programs. In doing this, effort should be made to support collaboration between multiple disciplines involved in pediatric pain rehabilitation (e.g., psychology, physical therapy, occupation therapy, pain medicine). It may be also beneficial to establish standard pain program protocols, such as the one presented by Maynard, Amari, Wieczorek, Christensen and Slifer [118]. The use of a uniform protocol across IIPT programs would also allow for further examination of the mechanisms within these programs that account for the significant functional improvements these patients experience.

Examining outcomes across levels of care will also be important. Future randomized controlled trials should focus on examining outcomes between intensive day treatment and inpatient treatment. Such a trial might allow researchers to better understand what treatment works for whom, and why. Examination of outcomes across treatment settings would allow for further examination of the most efficient and cost-effective way to deliver empirically supported treatment to children and their families. Lastly, continued examination of outcomes for e-Health is also warranted. In addition to providing services to patients in low-resource areas, mobile- and internet-delivered programs for pain management could be used to supplement in person treatments as patients complete more intensive rehabilitation services and transition back into social and academic environments.

## 5. Conclusions

In conclusion, chronic pain is a prevalent and persistent problem in childhood and adolescence. Rehabilitation for pediatric chronic pain is typically based on learning theory and on the biopsychosocial model of pain, which accounts for the complex interplay of the biological, psychological, social, and environmental factors that contribute to and maintain pain symptoms and related disability. Given all of the systems involved and effected by chronic pain, the treatment of chronic pain requires comprehensive treatment approaches, including psychological intervention, physical therapy, and occupational therapy. With the emergence of several targeted interventions to address the individual challenges each patient with chronic pain faces coupled with new means of overcoming barriers to access, the field is well-positioned to alleviate the suffering of youth with chronic pain and reduce their risk of transitioning to adults with chronic pain.

### Key Messages


Comprehensive multidisciplinary and interdisciplinary treatment based on behavioral medicine approaches are needed for children and adolescents with persistent pain.Pain Science Education is commonly implemented with several resources currently available, yet evidence for its use is scarce.Unique to pediatric rehabilitative approaches is the emphasis on including parents to optimize treatment outcomes.Innovative pain treatment intervention formats such as mobile applications and virtual reality enhance the delivery and reach of evidence-based tools.Comprehensive multidisciplinary/interdisciplinary treatment based on contemporary understanding of pain (neuro) science are needed for children and adolescents with persistent pain.


## Figures and Tables

**Figure 1 jcm-08-01267-f001:**
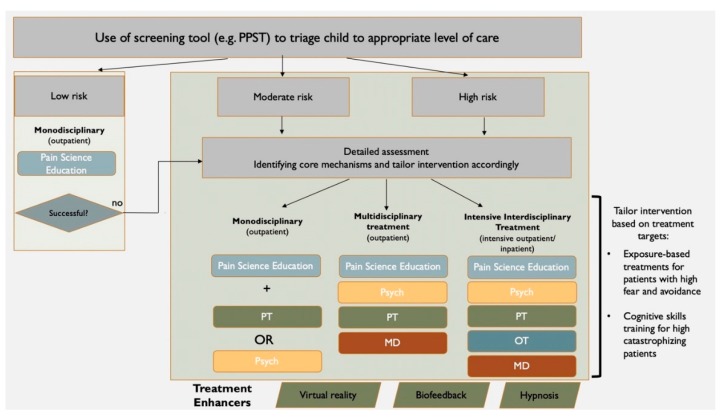
Schematic overview of Potential Future Directions for Clinical Practice. PPST = Pediatric Pain Screening Tool; PT = Physical Therapy; Psych = Psychology; MD = Medical Doctor; OT = Occupational Therapist.

**Table 1 jcm-08-01267-t001:** Best Evidence for Rehabilitation in Pediatric Chronic Pain.

	Evidence Supporting Interventions	Examples of Resources
**Individual Outpatient Interventions**		
**Pain Science Education**	Heathcote et al., 2019 ** [25]; Moseley & Butler [26], 2015; Pas et al., 2018 [27] *	Tame the Beast What is Pain? The Mysterious Science of Pain PNE4Kids A Journey to Learn About Pain
**Physiological Self-Regulation Training**	Eccleston et al., 2002 [28] **	
Biofeedback	Benore and Banez, 2013 [29]; McKenna et al., 2015 [30] *	Breathe2Relax BellyBio
Progressive Muscle Relaxation	Palermo, 2012 [10]	Progressive Muscle Relaxation Script
Self-Hypnosis	Liossi et al., 2003 [31]; Tome-Pires & Miro, 2012 [32];	
Guided Imagery	Van Tilburg et al., 2009 [33] *	
Mindfulness-based Stress Reduction (MBSR) and Yoga	Evans et al., 2010 [34]; Jastrowski Mano et al., 2013 [35] *	
**Cognitive Skills Training**	Eccleston et al., 2015 [36] **; Fisher et al., 2014 [14] **; Palermo et al., 2010 [15] **	
**Behavioral Exposure**	Kanstrup et al., 2017 [37]; Kemani et al., 2018 [24]; Wicksell et al., 2007 [38] *; Wicksell et al., 2009 [39]	
**Cognitive Behavioral Therapy for Insomnia (CBT-I)**	Palermo et al., 2017 [40] *	iSleep App CBT-I App
**Parent Coaching**	Eccleston et al., 2014 [36] **; Palermo, 2012 [10]	Conquering Your Child’s Chronic Pain Managing Your Child’s Chronic Pain When Your Child Hurts Pain in Children and Young Adults: The Journey Back to Normal
Problem-Solving Skills Training	Law et al., 2017 [41]; Palermo et al., 2016 [42] *	
**Multi-component Treatment Packages**		
Cognitive-Behavioral Therapy	Eccleston et al., 2014 [36] **; Fisher et al., 2014 [14] **; Palermo et al., 2010 [15] **	Cognitive-Behavioral Therapy for Chronic Pain in Children and Adolescents
Acceptance and Commitment Therapy	Pielech et al., 2017 [43]; Wicksell et al., 2009 [39]	Acceptance and Mindfulness Treatments for Children and Adolescents
**Physical Therapy**		
Strength and Endurance Training	Eccleston and Eccleston, 2004 [44]; Kempert et al., 2017b [45]; Mirek et al., 2019 [46]	
Gait and Posture Training		
**Occupational Therapy**		
Independence with Activities of Daily Living	Kempert et al., 2017a [47]; Kempert et al., 2017b [45]	
Desensitization	Sherry et al., 1999 [48]	
**Interdisciplinary Outpatient Pain Treatment**		
FIT Teens	Kashikar-Zuck et al., 2018 [49] *; Tran et al., 2016 [50] *	
2B Active	Dekker et al., 2016 [51]	
GET Living	GET Living, NCT: 03699007	
**Intensive Interdisciplinary Pain Treatment (IIPT)**	Hechler et al., 2015 [19] **	
**Emerging Pain Treatment Intervention Formats**		
One-day workshops	Coakley et al., 2018 [52]	The Comfort Ability
Internet and mobile applications	Bonnert et al., 2019 [53]; Palermo et al., 2018 [54]; Stinson et al., 2014 [55]	
Virtual Reality	Won et al., 2015 [56] *; Won et al., 2017 [57]	

Note: Tame the Beast and What is Pain? The Mysterious Science of Pain videos were not specifically developed for children. * denotes pilot studies; ** denotes systematic review and/or meta-analysis. All other studies listed are individual clinical trials or topical reviews.

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
