# Peer review of "Best-Evidence for the Rehabilitation of Chronic Pain Part 1: Pediatric Pain"

_jcm, 2019, doi:10.3390/jcm8091267_

Round 1

Reviewer 1 Report

Thank you for the opportunity to review this article entitled "Best-Evidence for Rehabilitation for Chronic Pain Part 1: Pediatric Pain." This manuscript reviews psychological and physical therapies typically used in the treatment of pediatric chronic pain. The manuscript is overall well written and well organized. I make some suggestions for improving the manuscript below:

Major:

Introduction

(1) The authors state on page 2 that they will not review pharmacological interventions because medications do not have high-quality evidence for management of pediatric chronic pain. Many of the interventions reviewed in this manuscript also do not evidence high-quality RCT evidence. Further, authors discuss use of MD level pain medicine in the future directions section. Suggest simply stating that this review is limited to psychological and behavioral interventions, but authors understand importance of MD pain management and reference to other reviews of pain medication in peds. Make this discussion consistent throughout manuscript.

(2) The value added by PT and exercise for muscle strengthening, balance, cardiovascular fitness, etc. is entirely missing from the introduction on page 3. Some of these physiological benefits are discussed on page 7. They should be discussed on page 3 as well.

Methods:

(2) The methods for this review are not clear. For instance, on page 2, what is considered "methodological sound clinical trials"? Please provide more standard reporting for review methods (e.g., inclusion criteria, exclusion criteria, age range, how many articles were reviewed and excluded, etc.).

(3) The search terms do not seem that they would target CBT, self-hypnosis, PT, OT. Are these the only search terms used? Can authors clarify search strategy?

(4) Unpublished data on page 10 should be deleted. Authors might reference published methods papers for such trials, but should state new trials are in progress rather than discuss unpublished results. 

Discussion:

(1) Include discussion of other pain screening tools (e.g., NIH PROMIS Pain interference scale,

Pediatric-modified Total Neuropathy Score) and temper discussion of the PPST, as PPST stratification has not yet been validated to predict treatment course/success. (Page 10-11)

(2) Related to above, while I applaud the authors for suggesting a treatment stratification model in Figure 1, no screening tools have been shown to be predictive of appropriate/best treatment. The authors should include a review of the limited treatment-failure literature to further guide this discussion and temper Figure 1 to not solely be based on PPST pain risk stratification.  

Minor:

(1) What does "cognition-targeted active approach to pain science" mean on Page 4, lines 124-125? Is this in reference to pain science education below? Please clarify this section.

(2) Several paragraphs are quite long and dense (e.g., Page 4 Pain science education; Page 6 Behavioral Exposure; Page 7 Parent coaching). Suggest breaking these sections in to smaller paragraphs.

(3) Addressing comorbidities section on Page 6 should likely go after the OT section. This seems out of place in the middle of the treatment review.

(4) The section on graded-in-vivo exposure is not clear (Page 8, lines 323-331). Should this go under behavioral exposure section or PT section?? Why are authors discussing the GET and 2B Active programs here? Please clarify this section.

(5) Please include more detail on cost saving (actual $ saved) on Page 9, line 382. This is important and helpful information to readers.

(6) Move "In sum" paragraph on page 10 to Conclusions section. This does not fit under Virtual Reality section.

(7) Occasional grammatical errors and words missing (page 6, line 237, "a pilot trial"?; page 12, line 490 “based on learning”). Please check manuscript carefully for small errors.    

Table 1:

(1) Left Justify information in columns and edit Table headers and formatting to make more readable. It is hard to digest information in Table 1 as presented.

(2) Suggest adding columns to Table 1 that (a) include pain population reviewed or studied (e.g., FM, JA, musculoskeletal, etc), (b) population age range, (c) brief methods (e.g., RCT, pre-post, review, meta-analysis), and (d) primary outcome (e.g., pain intensity, pain-related disability)

(3) Better explain and describe Table 1 in text. For instance, the descriptive sentence on page 3, line 110, states "intensive outpatient and inpatient treatment programs", but these headers are not used in the table and no mention about "Examples of Resources" or where these resources are coming from is discussed. Headers used in the Table should match the introductory text and sections reviewed on pages 4-9.    

Author Response

Please see the attachment for responses to reviewer 1's comments. 

Reviewer 2 Report

This is a very nice review article which carefully and thoughtfully addresses the broad spectrum of biopsychosocial issues in chronic paediatric pain rehabilitation.

I had four key thoughts

One

I noticed that the issue of neuromodulation is not mentioned as a possible intervention that may come up in the future. I understand that neuromodulation is more appropriate in adult chronic pain and that there are currently no data in children. However, the development of neuromodulation technologies is a phenomenon that is taking place. Since one of the aims of the review is to inform clinicians about emerging treatments this seems to be an omission. In this context it would seem appropriate to mention neuromodulation even if, it is not currently applicable to children. What form neuromodulation takes place in the future is unknown.

Two

The authors also explicitly stated that their frame of reference did not include medications and that medications are not be mentioned in the review. This is fine. However they also mention the comorbidity between chronic pain and psychiatric disorders. Whilst medication is outside the frame of this review, the complete omission of pharmacological treatments seems very odd, particularly when some of the comorbidities might need medication. In particular, in the section about co-morbid psychiatric diagnoses it might be useful to say something like:

Effective interventions—which may sometimes include the use of pharmacotherapy-- need to target co-morbid mental health 245 disorders and identify underlying mechanisms that serve to maintain mental health and pain 246 conditions.

In its current form the review reads almost as if the authors are completely anti-medication. Having treated many children/adolescent with chronic pain who were also extremely suicidal and where the risk of self-harm was very high, I think it is important to keep all options open. One never knows what the next clinical case may bring.

Three

After I had read the article it occurred to me that psychiatry was totally omitted from the review. I went back and checked to see if this was correct. I found this interesting since the Pain Team at our hospital works closely with psychiatry. They always pop in when one of their kids with chronic pain is suicidal or have a comorbidity that needs attention. The total omission of psychiatry is odd. Maybe the psychiatrists are subsumed under medicine?

Four

This is an association in the literature with early childhood adversity and chronic pain. In any serial referrals there is a small subset of kids who have experienced significant maltreatment. In these kids assessing risk and making sure they are safe before trying to implement other interventions—which will fail if the child is not safe—is a key issue. This type of risk is not mentioned anywhere in the article. Maybe it could be noted in the section about comorbidities. In clinical practice this is one of the most difficult scenarios to work with.

Other comments pertain to minor issues.

Line 162  and decreased muscle tension e.g., 41.

Having the number rather than the author names following the e.g. is disconcerting. If not journal style it should be changed. However from reading on it seems like this is journal style.

Line 169  This is the same issue. A number substituted for the author names. If not journal style then it is much easier for the reader to have the author names. This brings the distancing nature of academic writing to a new level (not even mentioning authors)!!

Line 490 “Rehabilitation for pediatric chronic pain is typically based learning theory on the biopsychosocial 490 model of pain,..

This line is not grammatical…needs to be fixed. 

Author Response

Please see the attachment for responses to reviewer 2 comments. 

Round 2

Reviewer 1 Report

Methods:

The below updated text was not in the manuscript version I received, but should be included: “However, to incorporate the most recent evidence, methodologically sound clinical trials (e.g., randomized controlled trials, sample size >20, clearly described intervention)”

The authors state the below information in the response letter. This should be added to the manuscript methods section. "The following search strategy was used: (((child OR pediatric) OR adolescent)) AND (chronic pain[Text Word]) AND rehabilitation[Text Word]). The inclusion of chronic pain AND rehabilitation resulted in a search that pulled for the primary interventions used in pediatric chronic pain rehab (i.e., CBT, PT, OT):"

Discussion:

The PSST does not have evidence of prognostic validity yet. Reference to “prognostic” factors, “prognostic” risk, etc. should be removed/edited on page 9. Page 11, Line 486, ‘may benefit from “including” graded exposure’. This language would better match the goal of this paragraph. The section on graded-in-vivo exposure (Page 9) was clarified in the author response but not the manuscript. More explanation of these programs is still needed to clarify this section in the manuscript.

Author Response

The below updated text was not in the manuscript version I received, but should be included: “However, to incorporate the most recent evidence, methodologically sound clinical trials (e.g., randomized controlled trials, sample size >20, clearly described intervention)”

Added this to the text. 

The authors state the below information in the response letter. This should be added to the manuscript methods section. "The following search strategy was used: (((child OR pediatric) OR adolescent)) AND (chronic pain[Text Word]) AND rehabilitation[Text Word]). The inclusion of chronic pain AND rehabilitation resulted in a search that pulled for the primary interventions used in pediatric chronic pain rehab (i.e., CBT, PT, OT):"

It is not required to include the search strategy, but we added this to the text. 

Discussion:

The PSST does not have evidence of prognostic validity yet. Reference to “prognostic” factors, “prognostic” risk, etc. should be removed/edited on page 9.

Removed "prognostic" from discussion of PPST. Edited sentence. 

Page 11, Line 486, ‘may benefit from “including” graded exposure’. This language would better match the goal of this paragraph.

Addressed. 

The section on graded-in-vivo exposure (Page 9) was clarified in the author response but not the manuscript. More explanation of these programs is still needed to clarify this section in the manuscript. 

The information included in the author response was to explain to you why we included a discussion of graded exposure where we did (i.e., in the behavioral exposure section as opposed to PT). We do not believe it makes sense and/or is necessary to include this clarification within the manuscript. Also, we do describe these programs (FIT Teens, GET Living, 2B Active) in an above section (please see Interdisciplinary Outpatient Pain Treatment; page 8). Re-describing them within the future directions section within the discussion feels redundent and not necessary.